# Aligning the Number of Parameters with the Number of Linear Regions for Improved Neural Network Approximation

**Alexey Podoprosvetov,**[*] **Vladimir Smolin,**[†] **Sergey Sokolov**[‡]
Keldysh Institute of Applied Mathematics,
Miusskaya sq., 4, Moscow, 125047, Russia
`llecxis@gmail.com, smolin@keldysh.ru, sokolsm@list.ru`

## Abstract

The paper addresses the "black box" problem of neural networks by analyzing the approximation properties of latent layers. It proposes that a key limitation preventing the practical achievement of universal approximation theorems is the mismatch between the growth rates of a network's parameters and the number of linear regions partitioning the input space. The question is examined how this imbalance is exacerbated in multidimensional cases, hindering effective learning. To resolve this, methods are suggested to align parameter counts with the number of linear regions, such as moving activations vectors to the surface of a hypercube, utilizing micro-columns, and leveraging the "blessing of dimensionality" in deep networks to decouple complex signals.

## 1 Introduction: Why Neural Network Approximation Mechanisms Remain Poorly Understood

A paradox persists: despite an abundance of textbooks, the inner workings of neural networks are still declared "incomprehensible" Barez et al. (2025). The root cause lies in the impossibility of visually interpreting transformations carried out in high-dimensional spaces. This is especially true of hidden layers: their activity is corrected by backpropagation, yet has no transparent meaning.

As a result, network optimization remains largely empirical, and numerous theories explain learning algorithms rather than the nature of the approximations that have already been formed.

We propose an approach based on topological analysis of the state spaces of latent-layer activation vectors. We examine one of the principal reasons for the discrepancy between practical training and the universal approximation theorem Cybenko (1989): the different growth rates of the number of network parameters and the number of linear regions partitioning the input signal space.

## 2 Survey of Recent Theories of Neural Network Approximation

Research into the organisation of knowledge in latent space is actively ongoing. Several directions can be identified:

(a) Application of geometric algebra enables formal investigation of spaces of arbitrary dimensionality without requiring their visual representation Sivgin et al. (2025).

(b) Study of sparse autoencoders (SAE) decomposes complex computations into thousands of human-interpretable concepts Transformer Circuits Thread (2025).

(c) Investigation of the "grokking" phenomenon — a sudden transition of the network from high error to accurate generalisation Power et al. (2022).

---

[*]ORCID: 0000-0002-3608-7895
[†]ORCID: 0000-0001-9030-6545
[‡]ORCID: 0000-0001-6923-2510

(d) Analysis of multi-layer interactions instead of individual neuron contributions He et al. (2024).

(e) Generalisation theory explaining why the found transformations succeed Dunefsky et al. (2024).

Each approach clarifies certain aspects of processes in the latent layers, but none describes their approximation properties as a whole. We propose complementing these methods with topological analysis of activation-vector state spaces, which allows broader conclusions about the nature of approximation.

## 3 VARIABLES IN THE BASIC FORMAL-NEURON NETWORK DESCRIPTION

Formal-neuron models, tracing back to McCulloch and Pitts McCulloch & Pitts (1943) and generalised by Widrow and Hoff Widrow & Hoff (1960), include the weight vector $\vec{W}_i^l = \{w_{ji}^l\}$, the *linear* activation $a_i^l$, and the nonlinear activation $o_i^l$. The weight vectors form a matrix $W^l = \{\vec{W}_i^l\}$ multiplied by the output of the preceding layer $\vec{O}^{l-1}$ (Fig. 1a):

$$\vec{A}^l = W^l \vec{O}^{l-1}; \quad \vec{A}^l = \{a_i^l = \vec{W}_i^l \, \vec{O}^{l-1}\}; \quad \vec{O}^l = \{o_i^l = \sigma(a_i^l)\}. \tag{1}$$

The nonlinear function $\sigma(x)$ can take various forms (Sigmoid, ReLU, SiLU, etc. Widrow & Hoff (1960)), including negative values and non-monotone dependencies.

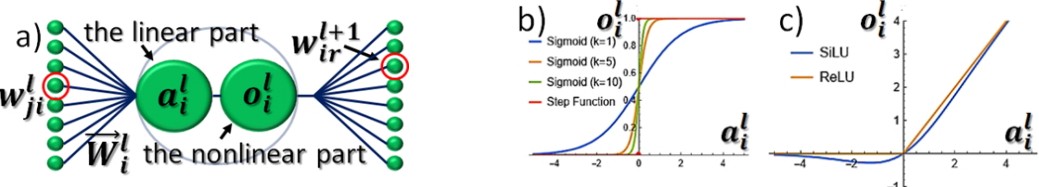

Figure 1: (a) Structure of formal neuron $i$ in layer $l$: input-connection weight vector $\vec{W}_i^l = \{w_{ji}^l\}$, linear activation $a_i^l$, and nonlinear activation $o_i^l$. (b) Nonlinear functions $o_i^l = \sigma(a_i^l)$: Sigmoid, $\sigma_{\text{Sigm}}(x) = 1/(1 + e^{-kx})$; (c) SiLU, $\sigma_{\text{SiLU}}(x) = x\,\sigma_{\text{Sigm}}(x)$, and ReLU, $\sigma_{\text{ReLU}}(x) = \max(0, x)$ Hammad (2024).

Network parameters $\vec{\theta} = \{w_{ji}^l\}$ are tuned by *gradient descent to minimise the loss function* $E = \sum_{q=1}^M E_q(\vec{\theta})$:

$$\Delta\vec{\theta}_q = -\alpha \sum_{q=1}^M \nabla E_q(\vec{\theta}); \qquad \nabla E_q(\vec{\theta}) = \left\{ \frac{\partial E_q}{\partial w_{ji}^l} \right\}. \tag{2}$$

Computing the partial derivatives $\frac{\partial E_q}{\partial w_{ji}^l}$ requires knowledge of $\frac{\partial E_q}{\partial a_i^l}$ for each layer. For the output layer these derivatives are determined directly by the approximation error; for hidden layers they are obtained via the backpropagation algorithm Rumelhart et al. (1986):

$$\delta_{iq}^l = \frac{\partial E_q}{\partial a_{iq}^l} = \frac{\partial E_q}{\partial o_{iq}^l} \frac{\partial o_{iq}^l}{\partial a_{iq}^l} = \left( \sum_{r=1}^N w_{ir}^{l+1} \delta_{rq}^{l+1} \right) \sigma'(a_{iq}^l); \qquad \frac{\partial E_q}{\partial w_{ji}^l} = \frac{\partial E}{\partial a_{iq}^l} \frac{\partial a_{iq}^l}{\partial w_{ji}^l} = \delta_{iq}^l \, o_{jq}^{l-1}, \tag{3}$$

where $\delta_{rq}^{l+1} = \frac{\partial E_q}{\partial a_{rq}^{l+1}}$, $\sigma'(a_{iq}^l) = \frac{d\sigma(a_{iq}^l)}{da_{iq}^l}$, and the index $q$ indicates dependence on the specific training pair $(\vec{X}_q, \vec{Y}_q^g)$. Individual values $\delta_{iq}^l$ (and correspondingly individual increments $\frac{\partial E_q}{\partial w_{ji}^l}$ from Eq. equation 2) are computed for each pair. Changes in the parameter vector $\Delta\vec{\theta}_q$ according to

Eq. equation 2 occur after a relatively randomly chosen subset ($M$ pairs) of the training set $(\vec{X}_q, \vec{Y}_q^g)$ is presented.

Pitfalls in deep learning are largely due to two factors. First, the need to reconcile in Eq. equation 2 the parameter increments composed of independently directed gradients $\nabla E_q$ for different $q$. Second, the finiteness of the learning steps. Infinitesimally small increments would ensure linearity of the influence of layer-by-layer changes, but training would take infinitely long. Choosing the largest feasible $\alpha$, on the other hand, leads to unequal influence of parameter changes at different network depths.

It is shown in Podoprosvetov et al. (2024) that gradient descent can in principle be driven by adjusting only the output layer; yet deep networks demonstrate the ability to solve complex problems. This means that stochastic gradient descent according to Eq. equation 2 is capable of supporting diverse optimisation methods implemented on multi-layer structures. Increasing the number of layers and using various regularisation methods raises the probability of forming successful approximations.

## 4 PRUNING

The flip side of the stochastic nature of neural network model formation is the presence of regions that are poorly tuned during training. Removing them from the structure — *pruning* — does not impair functioning and allows more efficient models to be created Cheng et al. (2024). This is one of the most dynamically developing directions in deep learning, having progressed from empirical methods to rigorous mathematical theories.

Modern methods such as PrunedLoRA propose dynamic elimination of less important components during fine-tuning, yielding more compact adapters compared with the standard LoRA technique Yu et al. (2025).

Applying pruning can reduce computational costs at inference and during the final stages of training by several times (sometimes by an order of magnitude). However, in the early stages considerable effort is expended on tuning parameters that will subsequently be discarded.

## 5 THE UNIVERSAL APPROXIMATION THEOREM — WHY IT DOES NOT WORK IN PRACTICE

In 1989, Cybenko proved the universal approximation theorem Cybenko (1989): a network with a single hidden layer and sigmoid nonlinearity can approximate any continuous function to any prescribed accuracy as the number of elements grows. However, the theorem only proved the *existence* of such settings, and gave no constructive path to achieving them. The theorem was subsequently extended to a broad class of nonlinear transformations $\sigma(x)$.

It was shown in Fokina & Oseledets (2019) that even knowing the exact weight values does not guarantee finding them via the standard SGD algorithm. The authors proposed a method of exponential error reduction by identifying regions with the largest error and adding new neurons at those locations. An alternative approach based on neural-network mapping that distributes the nonlinear properties of elements across the input signal space was proposed in Shen & Smolin (2025). Both methods reduce the approximation error for analytically defined functions, but the key property of Cybenko's theorem — the error tending to zero — is reproduced only for functions of a single variable. For multi-dimensional cases, zero error was not achieved in experiments.

The present article describes one of the main reasons for this discrepancy: without special measures or "luck" in the stochastic formation of the approximation, there are insufficient parameters for independently tuning all piecewise-linear regions of the input-space partition. The reason is the differing growth rates of the number of parameters and the number of regions as the number of hidden-layer elements increases.

## 6 MATRIX-VECTOR REPRESENTATION OF ACTIVATIONS AND WEIGHTS AND THEIR INTERRELATION

Before presenting the main idea, we consider certain properties of transformations in neural network elements. Figure 2 shows the structure and parameter notation of a deep network under a vector-matrix description of its operation.

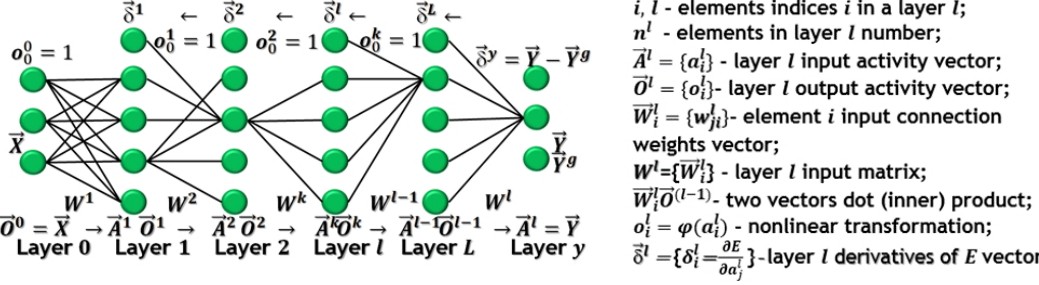

Figure 2: Structure of a deep neural network with $L$ fully connected hidden layers.

The geometric connection between the weight vectors $\vec{W}_i^l$ and the output activity vectors $\vec{O}^{l-1}$ of the preceding layer rests on their shared dimensionality, which allows the linear activation to be computed as a dot product: $a_i^l = \vec{W}_i^l \cdot \vec{O}^{(l-1)}$ (matrix $W^l = \{\vec{W}_i^l\}$, $\vec{A}^l = W^l \vec{O}^{l-1}$). The dimensionality of the "error" vector $\delta_{iq}^l$ (Eq. equation 3), which propagates back through the network, likewise coincides with the dimensionality of $\vec{A}^l$, enabling comparison of the changes $\Delta \vec{A}^l$ with the vectors $\delta_{iq}^l$. The coincidence of dimensionalities and the presence of dependencies between vectors (or their increments) of different natures permits a geometric interpretation of certain mutually dependent properties.

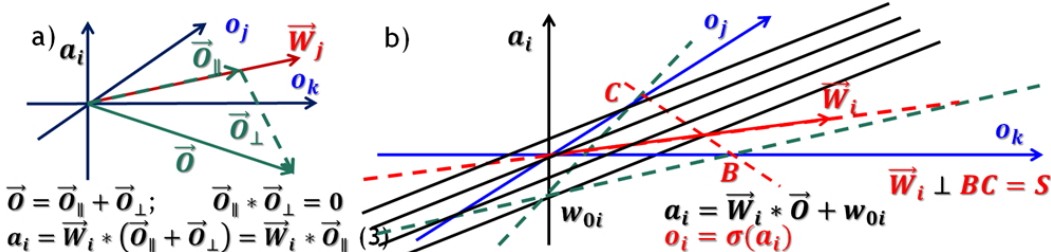

Figure 3: Geometric comparison of vectors $\vec{W}_i$ and $\vec{O}^{l-1}$ (indices $l$ and $l-1$ omitted): (a) the vector $\vec{O}$ can be decomposed into components $\vec{O}_\parallel$ and $\vec{O}_\perp$; (b) partition of the input signal space (output activity $\vec{O}^{l-1}$ of the preceding layer) into two half-spaces by the line $BC$.

The vector $\vec{O}^{l-1}$ can be expressed as the sum of two components: $\vec{O}_\parallel$ parallel to $\vec{W}_i$, and $\vec{O}_\perp$ perpendicular to it (Fig. 3a). Because the dot product of perpendicular vectors is zero, only the component $\vec{O}_\parallel$ contributes to the linear activation $a_i$. In a space of dimension $n_{l-1}$ — the number of elements in layer $l-1$ — $(n_{l-1} - 1)$ perpendiculars can be drawn to the vector. They form a hyperplane $S$ that divides the space of vectors $\vec{O}^{l-1}$ into two subspaces: on one side of the (hyper)linear surface (in the direction of $\vec{W}_i$), $a_i = \vec{W}_i \cdot \vec{O}_\parallel > 0$; on the other side, $a_i < 0$. On the surface $S$ itself, shown in Fig. 3b as line $BC$, $a_i = 0$. As the dimensionality $n$ of the space grows, the boundary separating it into two parts also increases its dimensionality, as illustrated in Fig. 4.

## 7 NON-LOCALITY OF NEURAL NETWORK APPROXIMATION

The realisable-state subspace $\hat{O} = \{\vec{O}_q\}$ is the set of all activation vectors that can be obtained when transforming input signals $\vec{X}$ from the approximated domain under fixed parameters $\vec{\theta}$. The

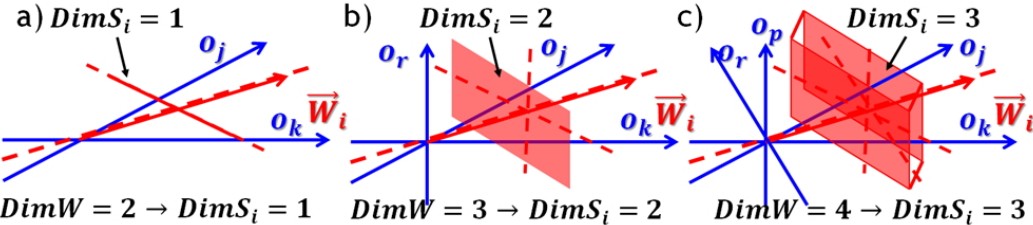

Figure 4: Growth of the dimensionality of subspaces $S_i : a_i = 0$, perpendicular to vectors $\vec{W}_i$, with increasing dimensionality of the spaces $W = \{\vec{W}_i\}$ and $O = \{\vec{O}\}$ (their dimensionalities are equal).

dimensionality of $\hat{O}$ is always less than that of the full space $O = \{\vec{O}\}$, but is generally greater than 1. When ReLU is used, the subspace $\hat{O}$ has a continuous piecewise-linear structure.

Each hyperplane $S_i$ (*the boundary where $a_i = 0$*), intersecting with $\hat{O}$, divides it into two parts. The characteristic sizes of these parts are comparable with the size of the entire subspace $\hat{O}$ and depend only on the orientation of the vectors $\vec{W}_i$. As the number of elements in the layers grows, these sizes do not tend to zero.

Nevertheless, the number of boundaries $S_i$ and the number of their intersections grows. Since the partition regions do not overlap, and the measure of the subspace $\hat{O}$ does not depend on the number of network elements (only the degree of its "crumpling" does), increasing the number of boundaries leads to smaller region sizes. Thus, subdivision of the realisable-state space into progressively finer regions is achieved not by locally reducing the scale of each boundary, but by globally increasing the number of their intersections — this is how the non-local character of neural network approximation manifests itself.

## 8   ONE-DIMENSIONAL SINGLE-LAYER APPROXIMATION

To understand multi-dimensional approximation, we first consider the case of a one-dimensional nonlinear function $y^g(x_1)$ (Fig. 5). The network contains one hidden layer; both the input and hidden layers also include an element with constant unit activity to implement biases. Network operation is described by:

$$a_i = \vec{W}_i^1 \vec{X}; \quad o_i = \text{ReLU}(a_i); \quad y = W^2 \vec{O}^1. \tag{4}$$

In Fig. 5 the function is approximated by three linear segments on the interval $[\min(x_1), \max(x_1)]$. Each segment is provided by its own ReLU function; the nonlinear activity of the second and third elements determines the positions of the kink points $p_1$ and $p_2$.

The first element has its kink point outside the considered interval, and its parameters $w_{01}^1$ and $w_{11}^1$ are tuned to match the middle segment. The parameters of the second element, $w_{02}^1$ and $w_{12}^1$, allow the activation point to be set at $x_1 = p_1$ and the slope to be chosen so that the sum with the first element matches the right segment. The third element is analogously tuned for the left segment.

Accuracy can be increased by adding hidden-layer elements, introducing new kink points and progressively improving the approximation. This procedure corresponds to the approach of Fokina & Oseledets (2019) and allows any prescribed error $\varepsilon$ to be achieved for any continuous function of one variable, demonstrating the fulfilment of Cybenko's theorem Cybenko (1989) in the one-dimensional case.

It is important to note that as the number of hidden-layer elements grows, not only the number of parameters but also the number of linear approximation segments grows proportionally. This is precisely what ensures the required accuracy. For multi-variable functions, as will be shown, the condition of matching the number of parameters to the number of linear regions may be violated, requiring either a fortuitous configuration during SGD or special adaptive algorithms.

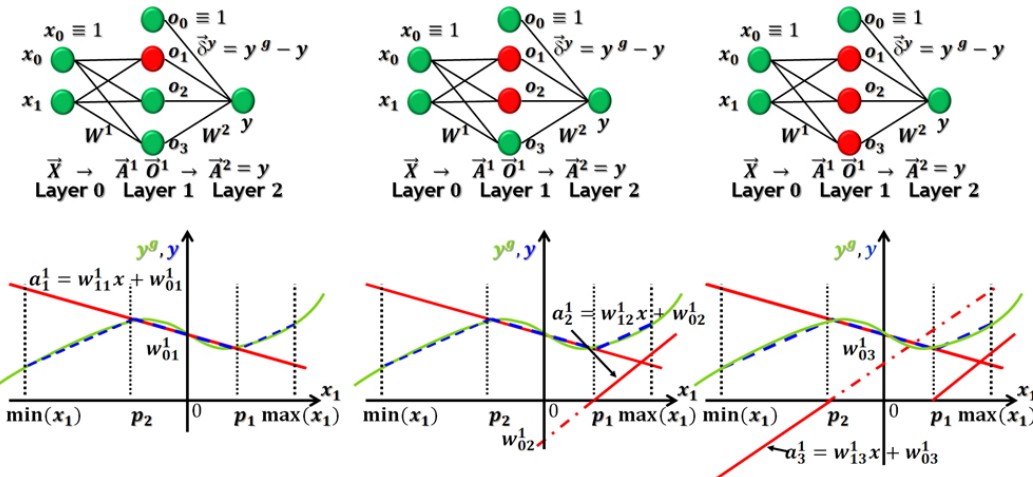

Figure 5: Piecewise-linear approximation of a function of one variable: the target function $y^g$ is approximated by successively activating three hidden-layer elements.

# 9    TWO-DIMENSIONAL SINGLE-LAYER APPROXIMATION

In the one-dimensional case, each additional boundary $S_i$ (point) divides one linear region into two. In the two-dimensional case the boundaries $S_i$ are straight lines of infinite extent, capable of crossing several regions of the subspace $\hat{O}$ simultaneously, creating several new partition regions.

Figure 6 shows the partition of the subspace by two element boundaries. They allow independent tuning of the properties of transformations in only two of the four delineated regions. Adding an element whose boundary does not intersect $\hat{O}$ creates no new regions but adds parameters for tuning. Adding elements with intersecting boundaries increases the number of regions faster than the number of parameters.

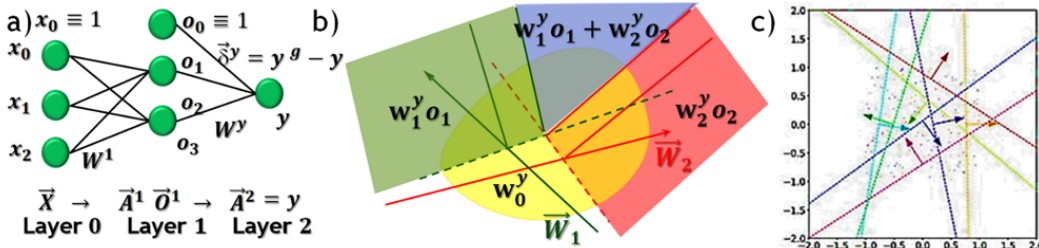

Figure 6: Piecewise-linear approximation of a function of two variables: (a) network structure; (b) piecewise-linear surfaces; (c) partition of the plane by eight boundaries $S_i$.

As shown in Fig. 7, the growth of the number of regions is quadratic in character (but depends not only on the number of neurons and the input dimensionality). Research Montúfar et al. (2014) shows that the maximum number of linear regions for a single-layer ReLU network, and the parameters required to describe them, can exceed the number of connection weights, which at a fixed parameter count limits the approximation capacity.

In general, when considering the two-dimensional partitioning of the subspace $\hat{O}$ into regions by boundaries $S_i$, as shown in Fig. 7, the growth of the number of regions is quadratic, but the dependence is not strict: by varying the positions of boundaries $S_i$ the number of regions can be both increased and decreased.

Exact approximation by a single-layer perceptron is achievable only for functions of a special form admitting variable separation:

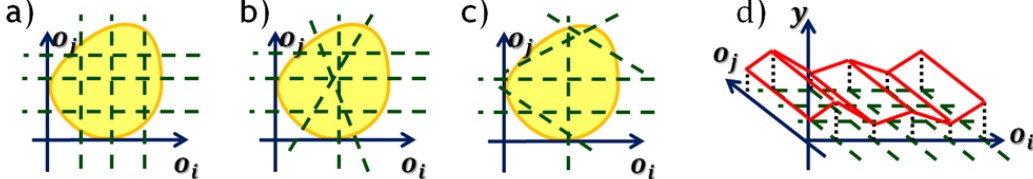

Figure 7: Piecewise-linear approximation of a function of two variables: (a)–(c) variants of partitioning the subspace $\hat{O}$ into regions by boundaries $S_i$; (d) one-dimensional approximation of a two-dimensional region.

$$y^g(\vec{X}) = \sum_{i=1}^{U} y^g(\mu_i^l) \quad \text{or} \quad y^g(\vec{X}) = \prod_{i=1}^{U} y^g(\mu_i^l), \quad \text{where} \quad \mu_i^l = \sum_{i=1}^{U} \beta_i^l o_i^l. \tag{5}$$

In this case the approximation problem reduces to the one-dimensional case (multiple one-dimensional dependencies, $U$ in total, can be approximated simultaneously and summed at the output layer). Approximation of general-form functions requires multi-layer networks.

## 10 NEURAL NETWORK MAPPING, ORTHOGONALISATION, AND CURVATURE CONTROL FOR APPROXIMATION PROBLEMS

Improving approximation accuracy need not rely solely on an excessive number of elements followed by pruning. An opposite approach is also possible: increasing the probability of achieving training results in which all parameters are used efficiently and pruning is not required. The present article is dedicated precisely to aligning the number of parameters with the number of linear approximation regions.

Other methods for increasing parameter efficiency also exist. Reference Podoprosvetov et al. (2024) describes approaches to distributing parameter participation in approximating different parts of the transformation via neural network mapping. Questions of orthogonalisation and curvature tuning of nonlinear transformations, which reduce the mutual influence of approximation regions, are addressed in Shen & Smolin (2025).

Achieving high approximation accuracy calls for the combined use of all these methods, including regularisation and normalisation, and not only the alignment problem addressed in this article.

## 11 TWO-DIMENSIONAL MULTI-LAYER APPROXIMATION

The boundaries $S_i$ are linear with respect to the variables of the preceding layer $\vec{O}^{(l-1)}$. However, due to the nonlinear transformations $o_i^{l-1} = \sigma(a_i^{l-1})$, these boundaries are no longer linear with respect to the input signal $\vec{X}$, especially in deep layers. Computer simulation allows the values of $\vec{X}$ to be identified at which the condition $a_i^l = 0$ holds for elements of different layers (Fig. 8a).

As expected, the shapes of the boundaries $S_i$ with respect to the input domain $\vec{X}$ turn out to be nonlinear; since ReLU was used, the nonlinearity takes the form of piecewise-linear broken curves.

As shown in Fig. 8b, the shape of the boundaries $S_i$ with respect to $\vec{X}$ is piecewise-linear. This is explained by the fact that the linear boundary (hyperplane) intersects the broken subspace $\hat{O}^{l-1}$, generating a one-dimensional broken curve. The kinks arise precisely at those values of $\vec{X}$ where $a_i^l = 0$ held in previous layers, i.e., at the intersections of preceding-level boundaries. Research shows that such cascaded boundary intersections do not lead to exponential growth of the number of linear regions with network depth, preserving a power-law dependence.

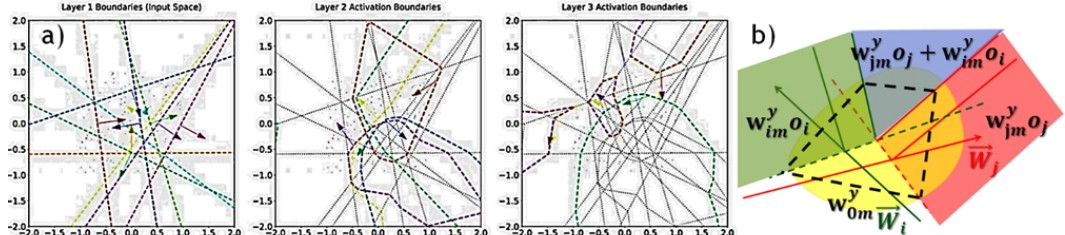

Figure 8: Piecewise-linear approximation of a function of two variables: (a) shapes of the partition of the subspace $\hat{X}$ into regions by boundaries $S_i$ in different layers; (b) intersection of the boundary $S_i$, linear with respect to the activity of the preceding layer $\vec{O}^{(l-1)}$, with the piecewise-linear shape of the subspace $\hat{O}^{(l-1)}$.

For three or more essential variables, visualisation becomes extremely difficult: the subspaces $\hat{O}^l$ have a three-dimensional broken shape and, intersecting with boundaries $S_i$, form two-dimensional broken boundaries with respect to $\hat{X}$, making their depiction on a plane largely uninformative.

## 12  DIMENSIONALITIES OF SUBSPACES $\hat{O}^l$ IN HIGH-DIMENSIONAL SPACES $O^l$

Since hidden layers may contain hundreds and thousands of elements, the dimensionality of the vectors $\vec{O}^l$, equal to their number of components, can be very large. Why should the realisable-state space $\hat{O}^l$ have lower dimensionality? Consider two cases:

(a) **A small number of input-signal components.** If the number of components of the input signal $X$ is small, then for any values of $X$ small changes in the corresponding vectors $\vec{O}^l$ can only occur in the directions of their increments under differentially small changes in the input signal components or any linear combination thereof:

$$d\vec{O}^l(\vec{X}) = \sum_{i=1}^{n_x} c_i \frac{\partial \vec{O}^l}{\partial x_i}\, dx_i; \qquad \frac{\partial \vec{O}^l}{\partial x_j} = \left\{ \frac{do_i^l}{dx_j}\, e_i^l \right\}. \tag{6}$$

The number of independent directions does not exceed $n_x$ and may become even smaller if certain directions degenerate under nonlinear transformations. Consequently, the dimensionality of the subspaces $\hat{O}^l$ for any layer $l$ cannot exceed the number of input signal components.

(b) **A large number of sensors but few degrees of freedom of the signal source.** If the input signal has many components (thousands or millions) but the observation targets objects with a small number of degrees of freedom, then the vectors $\vec{O}^l \in \hat{O}^l$ can only change in the directions of increments under changes in the variables describing those degrees of freedom. Consequently, regardless of the number of input-signal components, the dimensionality of $\hat{O}^l$ cannot exceed the number of degrees of freedom of the observed objects.

Experimental studies confirm that representations in deep networks are indeed concentrated on manifolds of substantially lower dimensionality than that of the layers themselves Ansuini et al. (2019).

## 13  GROWTH OF THE NUMBER OF PARAMETERS WITH THE NUMBER OF NETWORK ELEMENTS

In multi-layer perceptrons the number of connections $N_w$ grows proportionally to the square of the number of elements $N_e$:

$$N_w = \sum_{l=0}^{L} n^l n^{l+1}; \quad \text{for } n^1 = n^l = n^L: \quad N_w \sim (L-1)\left(\frac{N_e}{L}\right)^2, \quad \Rightarrow \quad N_w(N_e) \sim \frac{N_e^2}{L}. \quad (7)$$

At the same time, the growth of the number of regions when the dimensionality of the subspaces $\hat{O}^l$ exceeds two follows a power law corresponding to that dimensionality.

From estimate equation 7 it follows that the maximum number of parameters (connection weights) is achieved at $L = 2$. Yet it is well known that deep networks with hundreds and thousands of layers are effective for complex problems. What is the reason?

Our view: achieving high accuracy when approximating multi-dimensional transformations with a single-layer perceptron is impossible — the growth of the number of regions catastrophically outpaces the growth of the number of parameters. Deep networks, despite having fewer parameters for the same number of elements, possess properties that allow them to overcome this drawback.

## 14 THE BLESSING OF DEPTH AND DIMENSIONALITY

Bidirectional propagation of data and errors creates conditions for decomposing complex signals and identifying within them the essential variables that influence approximation accuracy. A large number of layers increases the probability of forming such transformations.

High dimensionality of the state vectors $\vec{O}^l \in \hat{O}^l$ creates conditions for forming a large number of mutually perpendicular regions in the subspace $\hat{O}^l$. This allows the shape of the subspace to be gradually changed, reducing the mutual influence of regions during adaptation and slowing the growth rate of their number with increasing $N_e$.

This phenomenon is known as the "blessing of dimensionality" — in contrast to the "curse of dimensionality" — and is actively studied in the current literature Kůrková (1997).

## 15 PATHS TO ALIGNING THE NUMBER OF PARAMETERS WITH THE NUMBER OF REGIONS

Since the growth of the number of piecewise-linear approximation regions is proportional to the number of hidden-layer elements raised to a power equal to the dimensionality of the state subspaces $\hat{O}^l$, and generally exceeds the quadratic growth of the number of parameters, achieving alignment requires finding ways to reduce the number of approximation regions. Several methods can be identified:

(a) **"Squeezing" the localisation of subspaces $\hat{O}^l$ onto the surface of the hyperspace $O^l$.**
Normalisation methods (e.g., BatchNorm Ioffe & Szegedy (2015)) compute the mean $\vec{M}^l$ and variance $\vec{D}^l$. When training signals are presented, each component $o_i^l$ is compared with $m_i^l$ and $d_i^l$, and the loss-function derivatives are corrected by the formulas:

$$\frac{\partial E_q}{\partial o_{iq}^l} = \begin{cases} \alpha(m_i^l - 2d_i^l - o_i^l), & o_i^l < m_i^l, \\ \alpha(m_i^l + 2d_i^l - o_i^l), & o_i^l \geq m_i^l. \end{cases} \quad (8)$$

This keeps activations near the surface of the hypercube, limiting region growth while keeping them away from the central part.

The computed values $\frac{\partial E_q}{\partial o_{iq}^l}$ allow calculation of $\frac{\partial E_q}{\partial a_{iq}^l}$ and $\frac{\partial E_q}{\partial w_{ji}^l}$ via Eq. equation 3, and then parameter increments $\Delta\vec{\theta}_q$ via Eq. equation 2.

Moving the realisable vectors $\vec{O}^l$ towards the hypercube boundaries creates conditions for them to reach vertex regions, where the infinite boundaries $S_i$ may intersect only a small portion of the hypercube containing the subspace $\hat{O}^l$. Fewer other boundaries $S_i$ are intersected, and a smaller number of regions is formed. For high-dimensional space the number

of hypercube vertices, equal to $2^l$ and already exceeding 1000 at $n^l = 10$, is practically unlimited for $n^l > 20$.

The parameter increments for hidden-layer matrices that shift activation vectors $\vec{O}^l$ towards hypercube surfaces can be combined in various proportions with increments computed by backpropagation, regularisation terms, and other parameter optimisation methods.

(b) Alignment can be achieved not only by reducing the number of piecewise-linear approximation regions, but also by increasing the number of parameters per network element. Nonlinear transformations $\sigma(a_i^l)$ can themselves have tunable parameters. Replacing ReLU with VReLU (V-shaped Rectified Linear Unit) with two independent ray-slope parameters slightly increases the total number of network parameters.

An effective approach is the formation of "micro-columns" — small groups of elements in each layer sharing a single output. The output of a "micro-column" is formed on the principle $\max_i a_i^l$ among group members. This approach not only multiplies the number of network parameters several times, but also allows localisation of the boundaries $S_i$ of the elements constituting the micro-column.

(c) Orthogonalisation of the arrangement of regions of the state subspace of the hidden-layer input signal $\hat{O}^l$ in the high-dimensional hyperspace $O^l$, discussed in Shen & Smolin (2025), has also been confirmed by experiments to reduce the excess number of piecewise-linear approximation regions.

(d) Decomposition of the transformation $\vec{X} \to \vec{Y}$ into components can yield the greatest effect by reducing the dimensionality of the components relative to the original transformation. However, this is a very broad topic and a detailed treatment of its many aspects lies beyond the scope of the present article.

## 16 CONCLUSIONS

This article has examined the problems of aligning the number of network parameters $N_w$ with the number of piecewise-linear approximation regions. The analysis shows that for successful neural network transformations, a decomposition of complex functions into low-dimensional components is necessary — this is an important path towards higher approximation accuracy.

The ReLU case considered here generalises readily to most other nonlinear functions, since practically all of them tend to linear asymptotes as the modulus of the argument grows Hammad (2024).

The importance of signal decomposition is confirmed by the successes of LLM and MLLM architectures OpenAI (2023), which exploit the decomposition of descriptions of a complex world into simple objects and actions already present in texts. However, a discussion of the problems of decomposing complex signals is a topic for a separate article.

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
